# Emerging Perspectives on the Rare Tubulopathy Dent Disease: Is Glomerular Damage a Direct Consequence of ClC-5 Dysfunction?

**DOI:** 10.3390/ijms24021313

**Published:** 2023-01-09

**Authors:** Giovanna Priante, Monica Ceol, Lisa Gianesello, Dario Bizzotto, Paola Braghetta, Lorenzo Arcangelo Calò, Dorella Del Prete, Franca Anglani

**Affiliations:** 1Kidney Histomorphology and Molecular Biology Laboratory, Nephrology Unit, Department of Medicine—DIMED, University of Padua, Via Giustiniani n° 2, 35128 Padua, Italy; 2Department of Molecular Medicine, University of Padua, 35121 Padua, Italy

**Keywords:** Dent disease, ClC-5, glomerulosclerosis, podocytes, CRISPR/Cas9 gene editing, nephrin, F-actin cytoskeleton

## Abstract

Dent disease (DD1) is a rare tubulopathy caused by mutations in the *CLCN5* gene. Glomerulosclerosis was recently reported in DD1 patients and ClC-5 protein was shown to be expressed in human podocytes. Nephrin and actin cytoskeleton play a key role for podocyte functions and podocyte endocytosis seems to be crucial for slit diaphragm regulation. The aim of this study was to analyze whether ClC-5 loss in podocytes might be a direct consequence of the glomerular damage in DD1 patients. Three DD1 kidney biopsies presenting focal global glomerulosclerosis and four control biopsies were analyzed by immunofluorescence (IF) for nephrin and podocalyxin, and by immunohistochemistry (IHC) for ClC-5. ClC-5 resulted as down-regulated in DD1 vs. control (CTRL) biopsies in both tubular and glomerular compartments (*p* < 0.01). A significant down-regulation of nephrin (*p* < 0.01) in DD1 vs. CTRL was demonstrated. CRISPR/Cas9 (Clustered Regularly Interspaced Short Palindromic Repeats/Caspase9) gene editing of *CLCN5* in conditionally immortalized human podocytes was used to obtain clones with the stop codon mutation p.(R34Efs*14). We showed that ClC-5 and nephrin expression, analyzed by quantitative Reverse Transcription/Polymerase Chain Reaction (qRT/PCR) and In-Cell Western (ICW), was significantly downregulated in mutant clones compared to the wild type ones. In addition, F-actin staining with fluorescent phalloidin revealed actin derangements. Our results indicate that ClC-5 loss might alter podocyte function either through cytoskeleton disorganization or through impairment of nephrin recycling.

## 1. Introduction

Dent disease is a rare nephropathy mainly caused by mutations in the *CLCN5* gene encoding the ClC-5 chloride channel that belongs to the voltage-gated chloride channel family (Dent disease type 1; MIM#300009). ClC-5 is mainly expressed in the kidney, particularly in the endocytic vesicles of proximal tubules (PT), where it participates to albumin and low-molecular-weight proteins (LMWP) endocytosis. ClC-5 contributes to the maintenance of intra-endosomal acidic pH and its C-terminus binds the actin-depolymerizing protein cofilin [1,2,3,4].

An important clue for the diagnosis of Dent disease type 1 (DD1) is a markedly increased level of urinary LMWP [5]. The cells mainly responsible for the uptake of filtered macromolecules are the proximal convoluted tubular epithelial cells (PTCs). Their apical plasma membranes form a brush border surface that facilitates LMWP receptor-mediated endocytosis from the lumen of the tubules via clathrin-coated pits. ClC-5 is also located at the cell surface of PTCs where it plays a role in the formation/function of the endocytic complex that involves megalin and the cubilin-amnionless complex (CUBAM) as receptors. Megalin and the CUBAM complex are the main actors in the albumin uptake mechanism since both megalin and cubilin have albumin binding sites in their structure [6,7].

A growing number of reports describe DD1 patients with nephrotic-range proteinuria and histological findings of focal segmental and/or global glomerulosclerosis [8,9,10,11,12]. A recent study showed that focal global glomerulosclerosis is common in DD1, particularly among older patients, and in those with lower eGFR [13]. In addition, foot process effacement of podocytes was frequently observed in the same patients, suggesting that podocytes may be involved in the glomerular pathology observed in DD1, with a role in the progression of renal failure [13]

Although receptor-mediated endocytosis by PT is a well known process [14], little information is available on endocytosis in the glomerular compartment. Evidence is emerging that podocytes are able to endocytose albumin using kinetics consistent with a receptor-mediated process [15]. Moreover, the preferential uptake by podocytes at the apical membrane, which faces the urinary space, would suggest that albumin uptake in these cells mirrors that by PT. This serves to recover albumin that leaks across the glomerular filtration barrier and enters the urinary space [16]. Human podocytes have also been shown to express megalin and cubilin both in vivo and in vitro [17,18]. Furthermore, our group revealed that ClC-5 is expressed in human glomeruli of normal and proteinuric kidneys, particularly in podocytes [19]. Interestingly, ClC-5 was found to be overexpressed in glomeruli of diabetic nephropathy and membranous glomerulonephritis at both the mRNA and protein levels, suggesting a role of this protein in albumin endocytosis by podocytes [19]. We also demonstrated that albumin overload increases ClC-5 expression in cultured human podocytes [20]. Beyond the mechanism of albumin endocytosis, immunogold-tracing electron microscopy and time-lapse fluorescent microscopy experiments suggest that podocytes endocytosis may regulate recycling of nephrin, attbuting to this mechanism an important role in podocyte slit diaphragm regulation [21].

It is well known that nephrin and Neph molecules are crucial for the maintenance of podocyte structure. The cell–cell adhesion module formed by these molecules not only serves as a guide for the formation and stability of podocyte foot processes, but also influences the podocyte intracellular signal transduction pathways [22]. The importance of nephrin for the development and function of glomerular podocytes and the slit diaphragm is testified by the alterations found in humans deficient in these molecules. Congenital nephrotic syndrome has been associated with mutations in the *NPHS1* gene encoding nephrin [23]. Nephrotic syndrome is characterized by an increased permeability of the glomerular filtration barrier for macromolecules and renal failure.

These data strongly support a possible role for ClC-5 in the maintenance of a proper glomerular structure and podocyte function, and suggest that the glomerular damage in DD1 might be a primary consequence of ClC-5 dysfunction.

In order to verify this hypothesis, we analyzed in vivo, in kidney biopsies from DD1 patients, and in vitro, in conditionally immortalized human podocytes, if ClC-5 loss of function could cause nephrin impairment and podocyte damage.

## 2. Results

### 2.1. Biopsies

Kidney biopsies from three DD1 patients carrying the two nonsense mutations p.(R34*), p.(Q600*) and the missense mutation p.(V308M) were analyzed to evaluate ClC-5, nephrin, and podocalixin expression. In these patients, Hematoxylin/Eosin staining revealed global glomerulosclerosis, although with different percentages (range 3.5–50), All the other glomeruli disclosed normal morphology. Transmission electron microscopy (TEM) analysis showed foot process effacement in the p.(Q600*) and p.(R34*) patients and normal podocyte structure in the p.(V308M) patient. Biopsies of patients carrying nonsense mutations showed chronic interstitial nephritis (Table 1).

By immunohistochemistry (IHC), 37 glomeruli from four control biopsies (CTRL) and 36 glomeruli from DD1 biopsies were evaluated. ClC-5 was down-regulated in DD1 vs. CTRL in both tubular and glomerular compartments (*p* < 0.01) (Appendix A).

Sixty-six glomeruli were paired between nephrin and podocalyxin immunofluorescences (IF) in CTRL and 32 in DD1. All CTRL glomeruli were positive for both markers, while nephrin was negative in 18% of DD1 glomeruli although they were all positive for podocalyxin, suggesting that the absence of nephrin was not due to podocyte loss. Quantitative morphometric analysis disclosed a significant down-regulation of nephrin in DD1 vs. CTRL and a significant increase of podocalyxin in DD1 vs. CTRL (*p* < 0.001 both) (Figure 1).

Since DD1 patients carried different *CLCN5* mutations, we analyzed whether nephrin and/or podocalyxin were differently expressed among these patients. We observed the highest down-regulation of nephrin in the patient carrying the p.(Q600*) mutation (*p* < 0.01), the highest down-regulation of podocalyxin in the patient carrying the p.(V308M) mutation (*p* < 0.01), and the lowest changes in both markers in the patient carrying the p.(R34*) mutation (Appendix A).

### 2.2. Generation of CLCN5 Mutant Human Podocyte Cells through CRISPR/Cas9 Genome Editing

To understand whether *CLCN5* mutations could be a direct cause of nephrin downregulation and of the podocyte damage observed in the glomerular compartment of DD1 patients, we edited the *CLCN5* gene in conditionally immortalized human podocyte cells by CRISPR/Cas9 technology. We obtained two different clones carrying the homozygous c.100delC mutation in exon 2 both at DNA and mRNA level leading to the premature stop codon p.(R34Efs*14) (Appendix A).

Using real-time PCR (qRT-PCR) and In-Cell Western (ICW), relative levels of *CLCN5* mRNA and ClC-5 protein respectively were examined in mutant versus WT clones (clones subjected to CRISPR/Cas9 editing but without mutation). As shown in Figure 2, we demonstrated that mutant clones (*CLCN5*^−/−^) showed a significant downregulation of ClC-5 at both mRNA (Figure 2A) and protein levels (Figure 2C) (*p* < 0.001 both). Specifically, gene editing resulted in an approximately 90% and 82% reduction of *CLCN5* transcript with respect to WT in *CLCN5*^−/−^ clone 1 and *CLCN5*^−/−^ clone 2, respectively (Figure 2B). At protein level, we found an approximately 50% and 30% reduction of ClC-5 in *CLCN5*^−/−^ clone 1 and *CLCN5*^−/−^ clone 2, respectively (Figure 2D).

### 2.3. Qualitative Analysis of CLCN5 mRNA in Mutant Clones

To understand why a very precocious stop-codon mutation at codon 48 could result in ClC-5 protein production—albeit at low level—we explored two possibilities: (1) that exon 2 skipping could encompass the mutation thus enabling protein synthesis, (2) that a ClC-5 isoform lacking the 5′ region (from exon 1 to 6) of the transcript, isoform *CLCN5*-206, could be upregulated.

By amplifying *CLCN5* mRNA using primers encompassing exon 2 of the canonical isoform (*CLCN5*-204), we evaluated the presence of an alternative spliced *CLCN5* transcript. We did not identify smaller PCR products than expected in mutant clones compared with WT clones, thus indicating that no alteration in mRNA processing occurred (Figure 3).

*CLCN5*-206 isoform was amplified using a nested PCR because of the low amount of this isoform in the human kidney. We first analyzed the expression of the isoform in different human kidney cell lines: HK2 (proximal tubular cells), HMC (mesangial cells), HUVEC (endothelial cells), podocytes, and glomerular parietal epithelial cells (PEC). Total RNA from a human kidney biopsy was also analyzed. Fibroblast cells were used as a negative control. Although nested PCR cannot be quantitative for its intrinsic nature, we tried to give a measure of *CLCN5*-206 expression by judging band intensity from the Agilent pattern of amplicons. The PCR products for the housekeeping gene, obtained from the first round of nested PCR, assured us that the starting conditions for the second round of PCR were quite similar for all samples. HK2, HMC, and renal biopsy showed the highest amplicon signal, fibroblasts and HUVEC did not show this isoform, podocytes and PECs exhibited the lowest signal (Figure 4A). With the same criteria, we examined *CLCN5*-206 isoform expression in mutant versus WT clones, and found that mutant clones had less amplicon signal of this isoform than WT ones (Figure 4B).

### 2.4. Effects of CLCN5 Downregulation on Human Podocyte Cells

Nephrin expression, both at the mRNA and protein levels, was analyzed in the two homozygous mutant clones (*CLCN5*^−/−^) versus WT clones. Cubilin, a partner of ClC-5 in receptor-mediated endocytosis, was also evaluated. Actin cytoskeleton, which is known to interact with nephrin for assuring a functioning slit diaphragm, was analyzed using phalloidin fluorescent staining. These evaluations were performed in basal conditions and following albumin treatment.

#### 2.4.1. CLCN5 Downregulation Alters Nephrin Expression

As shown in Figure 5, mutant clones showed a significant downregulation of nephrin at both mRNA (Figure 5A) (*p* < 0.05) and protein levels (Figure 5C) (*p* < 0.001). The downregulation of the *CLCN5* gene caused an approximately 50% and 70% reduction of nephrin transcript compared to WT in *CLCN5*^−/−^ clone 1 and *CLCN5*^−/−^ clone 2, respectively (Figure 5B). At the protein level, we found about 60% and 45% reduction versus WT in *CLCN5*^−/−^ clone 1 and *CLCN5*^−/−^ clone 2, respectively (D).

Regarding cubilin, we found no difference in the expression both at mRNA and protein level between WT and mutant clones (Appendix A).

#### 2.4.2. CLCN5 Downregulation Alters Podocyte Actin Cytoskeleton

We next investigated for cytoskeletal abnormalities. We assessed the distribution of actin fibers in WT and mutant clones. Podocyte cells not subjected to CRISP/Cas9 were also analyzed (human podocytes). Human podocytes and WT clones showed a pattern of F-actin filaments distributed as bundles of stress fibers along the cell axis, arranged neatly and unbranched, as visualized by fluorescence microscopy. In mutant clones, the orderly arranged stress fibers of the podocyte actin cytoskeleton were disrupted and a marked redistribution of F-actin fibers toward the periphery was observed (Figure 6A). The redistribution of stress fibers toward the periphery could also be interpreted as a decrease of actin expression, since the most of the cytoplasm turned out to be devoid of F-actin.

We also analyzed if there were differences in actin cytoskeleton among human podocytes, heterozygous (*CLCN5*^+/−^), and homozygous (*CLCN5*^−/−^) mutant clones. As shown in Figure 6B, a difference in the F-actin pattern was observed: the degree of actin derangement due to *CLCN5* mutation appeared to be gene dosage-dependent.

### 2.5. Effects of CLCN5 Downregulation on Human Podocyte Cells Treated with Albumin

In order to analyze the effect of ClC-5 downregulation on the podocyte response to albumin overload, we incubated mutant and WT clones with 10 and 30 mg/mL of albumin at different time points (24, 48, and 72 h) for mRNA and protein analysis.

To rule out the detrimental effect of albumin overload on podocytes, we estimated by methylene blue assay the number of living cells at 48 and 72 h after incubation with BSA. At both 48 and 72 h, there was a significant decreasing of cell number with increasing of BSA concentrations in WT and mutant clones (Figure 7).

#### 2.5.1. Albumin Modulates ClC-5, Nephrin and Cubilin Expression

We first examined albumin effects on ClC-5 expression. While at the mRNA level, there was a clear downregulation of the *CLCN*5 transcript (Figure 8A), at the protein level, we detected an upregulation of ClC-5 protein both in WT and mutant clones. ClC-5 increased directly with the increase of albumin concentration, although significant only at 48 h for WT clones (*p* < 0.005) and at 72 h (*p* < 0.05) for mutant clones (Figure 8B). The significantly lower level of *CLCN*5 in mutant clones in respect to WT was confirmed at the mRNA level at each concentration and time tested (*p* < 0.001) (Figure 8A). At the protein level, instead, this difference was significant only at 48 h (*p* < 0.05) due to the upregulation of ClC-5 in mutant clones (Figure 8B).

To rule out whether the up-regulation of ClC-5 in mutant clones was due to the up-regulation of ClC-3 and ClC-4—that share about 70% of amino acid similarity with ClC-5—we quantified all these proteins by ICW in the course of albumin treatment. We found that ClC-3 was upregulated both in WT and mutant clones in a dose-dependent manner (*p* < 0.05) (Figure 8C), while ClC-4 was barely expressed, both at basal and stimulated conditions (Figure 8D).

We also examined whether ClC-5 upregulation in mutant clones during albumin treatment was caused by the upregulation of the *CLCN5*-206 isoform. Although the RT/PCR approach we used to detect this isoform was not quantitative, we observed in the mutant clones after 48 h of exposure to 30 mg/mL albumin that the amplicon corresponding to isoform 206 was more evident than in the control condition. In WT clones, the albumin treatment seemed to decrease the expression of this isoform (Figure 9).

With albumin treatment, the mutant clones showed significant upregulation of nephrin only at 48 h. Nevertheless, nephrin levels remained at a significantly lower expression than WT clones, except after chronic exposure (30 mg/mL of albumin at 48 h), as the upregulation of nephrin in mutant clones brought nephrin to comparable level of WT clones (Figure 10A). The effect of a chronic exposure was also evident at mRNA level both in mutant and WT clones. In fact, nephrin transcripts were significantly higher after 48 h of exposure to 30 mg/mL albumin (Figure 10B). Notably, in WT clones, the significant upregulation of nephrin at the mRNA level was not reflected in an upregulation at the protein level.

A significant upregulation in *CUBN* mRNA expression at the highest albumin concentration was found in WT and mutant clones after 24 h (*p* < 0.01) (Figure 11A). This pattern was confirmed at the protein level at 72 h (*p* < 0.005) only in WT clones (Figure 11B). No differences were detected between WT and mutant clones in each experimental condition.

#### 2.5.2. Albumin Alters Podocyte Actin Cytoskeleton

Derangement of F-actin was confirmed in mutant clones after albumin treatment. Albumin treatment did not further modify the severe actin derangement already evident in basal conditions. Altered morphology and severe loss of stress fibers were also observed in WT clones as well as in human podocyte treated with albumin at 30 mg/mL for 72 h (Figure 12). The degree of the alteration observed in mutant clones in basal conditions was quite similar to normal podocytes after albumin chronic exposure.

## 3. Discussion

The role of podocytes in glomerular filtering and albumin handling has been at the heart of glomerular research for decades. Podocytes have a prominent role in maintaining the integrity of the glomerular filtration barrier, and their injury, either for genetic or environmental causes, is known to be the major reason of marked albuminuria and nephrotic syndrome [24]. The molecular makeup of podocytes was uncovered, and led to the understanding of the complexity of podocyte functions. Nephrin is one of the crucial components of the slit diaphragm and podocalyxin is a constituent of the glycocalyx of podocytes. We demonstrated that podocytes express ClC-5 both in vivo, in human kidney biopsies [19], and in vitro, in an immortalized human podocyte cell line, in which we showed that albumin upregulated ClC-5 expression. [20]. The function of ClC-5 in PTCs is well established. What is still unknown is its role in glomeruli. DD1 might be a good human model for studying ClC-5 function in podocytes. Based on the knowledge that in patients with DD1, clinical presentation with heavy proteinuria, even in the range of nephrotic proteinuria, is not so uncommon [25], we hypothesized that podocyte impairment may be the origin of this clinical feature.

The results we obtained in renal biopsies of three DD1 patients seem to support this hypothesis, disclosing for the first time a significant downregulation of nephrinin glomeruli of DD1 patients, suggesting a role for ClC-5 in nephrin expression. The patients carried different types of mutations, two stop codons, and one missense. The p.(Q600*) nonsense mutation affects the ClC-5 CBS1 domain in the C-terminus of the protein. This variant seemed to be the most dangerous one for the slit diaphragm, leading to the highest down-regulation in nephrin expression without changes in podocyte number. These data were further supported by the foot process effacement observed by TEM. The p.(R34*) seemed to be the less-damaging mutation, despite its being predicted to produce a very precociously truncated non-functional protein. Premature stop codons like this may occasionally be bypassed when translational read-through allows the decoding of stop codons as sense codons, thus enabling protein translation [26]; alternatively they can prompt exon skipping by altering ESE and ESS motifs [27]. Both hypotheses can explain this phenotype.

Consistent with our finding in DD1 glomeruli, we demonstrated for the first time that in human podocytes lacking ClC-5, nephrin was downregulated and the actin cytoskeleton was altered.

We used CRISPR/Cas9 technology to obtain clones carrying the very premature stop codon mutation p.(R34Efs*14). We demonstrated that this mutation was able to reduce ClC-5 expression both at mRNA and protein level although with different efficiency. At the mRNA level, ClC-5 downregulation was remarkable. It means that degradation of the transcripts containing the premature stop codon was activated via the nonsense-mediated decay (NMD) pathway [28]. What was surprising was the detection of ClC-5 protein in the mutant clones, and its upregulation during the albumin treatment despite the presence of a clear downregulation at the mRNA level. Furthermore, the mutant protein should have been completely undetectable due to the lack of about the 93% of the aa chain, and because the ClC-5 antibody we used could only recognized an epitope at the C-terminus of the protein. We tried to understand this phenomenon and were able to exclude exon skipping, while we cannot exclude a stop codon read-through (RT). However, RT in human cells was rarely shown, and the stop codon UAA, which was generated by the c.100delC mutation, has the lowest RT potential [26].

Having shown that the ClC-3 chloride channel was expressed in human podocytes and upregulated by albumin, we reasoned that the ClC-3 that shares 70% of similarity with ClC-5 could have accounted for the ClC-5-positive signal detected in mutant clones in basal conditions and during albumin treatment.

The short isoform *CLCN5*-206, which lacks the first six exons where the mutation is located, was found to be expressed in podocytes, albeit at a very low level, and upregulated by albumin in mutant clones. However, this isoform, for which the functional significance is still unknown, was shown to be barely expressed in the kidney. To look for it, we used a nested PCR starting from 200 ng of RNA, because it was undetectable with a normal RT/PCR. Thus, we argued that the ClC-5-positive signal detected during albumin treatment was mainly due to the upregulation of ClC-3. ClC-3 and ClC-4, like ClC-5, function as Cl^−^/H^+^-exchangers. As vesicular ClCs, they are believed to assist the acidification of intracellular vesicles by electrically shunting the currents of the vesicular H^+^-ATPase. ClC-3 may form heteromers with ClC-4 and ClC-5. Due to the strict structural similarity and biological function, it may be reasonable to suppose that ClC-3 can vicariate the ClC-5 function when ClC-5 is missing, thus explaining its upregulation during albumin treatment in our in vitro model. A role for ClC-3 in endocytosis, however, has not yet been demonstrated. Although ClC-3 is prominently expressed in apical endosomes of renal proximal tubules, proximal tubular endocytosis of luminal proteins was not affected in *Clcn3*^−/−^ mice, in contrast with the strong reduction of proximal tubular endocytosis upon disruption of ClC-5 [29]. Thus, we assumed that ClC-3 in our model was unable to compensate for the lack of ClC-5.

Once we were confident that ClC-5 function was lost in our in vitro model, we looked for nephrin expression and showed that loss of ClC-5 in podocytes led to downregulation of nephrin in mutant clones both at the mRNA and protein levels.

Our results on F-actin staining of mutant podocytes indicate that ClC-5 loss also causes phenotypic changes consisting in actin rearrangement that was shown to be similar to that induced by chronic exposure to albumin in WT clones. Morigi et al. already demonstrated that exposure to albumin induces cytoskeleton rearrangement and dedifferentiation of podocytes [30]. With our experiments, we showed that not only albumin, but also loss of ClC-5, caused remodeling of the podocyte actin cytoskeleton.

Podocytes are unique epithelial cells with a large cell body and long processes. Microtubules and intermediate filaments form the framework of the podocyte cell body and primary processes, whereas the secondary processes are rich in actin. The complex and unique actin cytoskeleton structure enables podocyte to adapt to fluctuating pressures and potentially harmful molecules contained in the primary filtrate [31]. Accordingly, remodeling of the actin cytoskeleton is closely related to foot process effacement, podocyte loss, and proteinuria.

Actin-based endocytosis has now emerged as a regulator of podocyte integrity. Dynamic actin filaments are crucial components of clathrin-mediated endocytosis, since the actin cytoskeleton is thought to provide force for membrane invagination or vesicle scission. Evidence in PTCs suggests that the large intracellular ClC-5 C-terminus plays a crucial function in mediating the assembly, stabilization, and disassembly of the endocytic complex via protein–protein interactions. The Hryciw group demonstrated that the C-terminus of ClC-5 binds the actin-depolymerizing protein cofilin [32]. The forming of the nascent endosome seems to require recruitment of cofilin by ClC-5 to localize the dissolution of the actin cytoskeleton, thereby allowing the endosome to pass into the cytoplasm.

The results we obtained on the *CLCN5* gene-dosage effect on podocyte cytoskeleton derangement suggest that even in human podocytes ClC-5 might interact with the actin fibres to enable endocytosis. As a matter of fact, a defective transferrin endocytosis has been recently demonstrated in cultured podocytes with genetic knockdown of *CLCN5* gene by Solanki et al. [33]. Since podocyte endocytosis has a critical role in the internalization and recycling of nephrin [16], we argued that nephrin depletion in mutant clones might be caused by alteration of podocyte endocytosis due to ClC-5 loss.

From our results, we could infer that the loss of ClC-5 function in human podocytes is likely to alter podocyte function both directly, through cytoskeleton derangement, and indirectly, through downregulation of nephrin due to impairment of nephrin recycling.

In conclusion, the results of our study reveal that glomerulopathy in DD1 patients might be a direct consequence of ClC-5 impairment in podocytes, and that Dent disease type 1 should be considered not only as a tubulopathy, but also as a podocytopathy.

## 4. Materials and Methods

### 4.1. Renal Biopsies

Kidney biopsy specimens from three DD1 patients had been performed in the suspicion of glomerulopathy due to the presence of proteinuria. All biopsies were performed for diagnostic purposes and available for immunolabeling studies subject to informed consent. DNA analysis previously performed for diagnostic purpose revealed they carried three different *CLCN5* mutations i.e., two nonsense mutations p.(R34*), p.(Q600*) and a missense mutation p.(V308M) (Table 1).

Four control cortical tissues were obtained from nephrectomies for renal cancer (sites remote from the tumor-bearing renal tissue), disclosing a normal morphology and no positivity in immunofluorescence.

The study was approved by Padua University Hospital’s Ethical Committee, protocol 0007452 (1 February 2018).

#### 4.1.1. Immunohistochemistry

In order to detect ClC-5, immunohistochemistry (IHC) was conducted on formalin-fixed, paraffin-embedded sections using an indirect immunoperoxidase method. Specimens were treated as previously described [19], and incubated overnight with rabbit anti-human ClC-5 in PBS at 4 °C in a humidified chamber (Table 2). Immunolabeling specificity was confirmed by incubating without any primary antibody. Images were acquired using the Diaplan light microscope (Leitz, Como, Italy) and a 20×/0.45 objective using a Micropublisher 5.0 RTV camera (Teledyne QImaging, Surrey, BC, Canada).

#### 4.1.2. Immunofluorescence

To detect nephrin and podocalyxin, immunofluorescence (IF) analyses were performed on serial sections of the same biopsies evaluated by IHC. Samples were treated as previously described [20] and incubated overnight with the appropriate primary antibody diluted in PBS 5% BSA at 4 °C (Table 2). Sections were then incubated with the appropriate fluorescent secondary antibody diluted in PBS 5% BSA at room temperature (Table 2) [20]. Nuclei were counterstained with 4′,6-diamidino-2-phenylindole (DAPI, Vector Laboratories, Burlingame, CA, USA) diluted 1:1000 in PBS. Negative controls were run by omitting primary antibody. Images were acquired with a DMI6000CS-TCS SP8 fluorescence microscope (Leica Microsystems, Wetzlar, Germany) with a 20×/0.4 objective using a DFC365FX camera (Leica Microsystems, Wetzlar, Germany) and analyzed with the LAS-AF software 3.1.1 (Leica Microsystems, Wetzlar, Germany).

#### 4.1.3. Morphometric Analysis

IHC signal for ClC-5 and IF signals for nephrin and podocalyxin were quantified by morphometric analysis using the Image-Pro Plus 7.0 Software (Media Cybernetics, Abingdon, UK). IHC signals were acquired for all images with the same brightness and contrast, IF signals with the same time exposure, gain, and intensity. In each biopsy, cortical tubular interstitial tissue was separately evaluated from glomeruli. Quantities were expressed as the mean area covered by pixels (%) [34].

### 4.2. Creation of CLCN5 Mutant Podocyte Cell Clones

#### 4.2.1. Human Podocytes

Human podocytes were kindly provided by Prof. Saleem and maintained in RPMI 1640 medium supplemented with 10% foetal bovine serum (FBS; F7524, Sigma-Aldrich, St. Louis, MO, USA), Insulin-Transferrin-Selenium supplement (ITS; Sigma-Aldrich, St. Louis, MO, USA), 2 mM L-glutamine, and antibiotic mixture, as previously reported [35]. To stimulate cell proliferation, podocytes were cultivated at 33 °C in 5% CO_2_ (permissive conditions). To induce differentiation, they were maintained at 37 °C in 5% CO_2_ (non-permissive conditions) for at least 2 weeks, and verified to be free of mycoplasma contamination through the N-GARDE Mycoplasma PCR Detection kit (EuroClone S.p.A, Milan, Italy). Cell density was kept below 90% to allow differentiation. Cells were used between passages 7 and 12. To test albumin effects, cells were cultured in serum deprivation (1%) starting from 24 h before stimulation.

#### 4.2.2. CRISPR/Cas9-Mediated Genome Editing

CRISPR/Cas9 gene editing of *CLCN5* in conditionally immortalized human podocytes was used to obtain clones with an early stop codon mutation.

Guide RNA (sgRNA AGACCGGGATAGGCAC**C**GAG(AGG)) was designed in the exon 2 using the *Breaking-Cas* web server, freely accessible on-line at: http://bioinfogp.cnb.csic.es/tools/breakingcas (accessed on 22 November 2022) [36]. Podocyte cells were transfected with RNP complexes using the TransIT-CRISPR^®^ reagent (#T1706, Sigma Aldrich, St. Louis, MO, USA) following manufacturer’s protocol. Briefly, RNPs were prepared by mixing Cas9 protein (IDT Integrated DNA Technologies, Coralville, IA, USA) and in vitro transcribed sgRNA. RNPs were mixed with the transfecting agent and then cells plated at 80% confluence were transfected. 48 h after transfection, cells were detached and seeded into a 96-well plate via serial dilution to obtain isolation of single-cell clones. Cells were incubated in a 33 °C, 5% CO_2_ incubator, as above.

After the establishment of single-cell clones, they were expanded and characterized in order to identify mutant clones. Individual clones were analyzed for purity and genotype characterization by PCR and Sanger sequencing. Out of 72 clones sequenced by Sanger, five carried the c.100delC, p.(R34Efs*14) heterozygous mutation. In order to obtain clones carrying the mutation in homozygosity, the procedure was repeated on two heterozygous clones. Out of forty clones obtained after this second round of CRISPR/Cas9 editing, two were found to carry the mutation in homozygosity.

These two clones (*CLCN5*^−/−^ 1 and 2) and three clones with normal genotype as control (clones subjected to CRISPR/Cas9 editing but without mutation: WT 1, 2, and 3) were plated and cultured at 37 °C in 5% CO_2_ (non-permissive conditions) for at least 2 weeks to induce differentiation and for successive experimental treatments. At this time, the clones were again characterized by PCR and Sanger sequencing to ensure that they maintained the original genotype both at DNA and mRNA levels. For evaluating the basal expression of ClC-5 and Nephrin, two independent experiments were performed. In each experiment, clones were plated in triplicate on 6 or 96-well plates for RT-PCR and In Cell Western (ICW) analysis, respectively.

#### 4.2.3. CLCN5 Mutation Analysis

##### High Resolution Melting (HRM) Analysis

Genomic DNA was extracted using the NucleoSpin DNA RapidLyse (Macherey-Nagel GmbH & Co. KG, Düren, Germany) according to the manufacturer’s instructions. 30 ng of DNA, quantified with the NanodropONE spectrophotometer (Thermo Fisher Scientific, Waltham, MA, USA), were amplified in a final reaction volume of 25 μL containing 10 mM pH 8.3 Tris-HCl, 50 mM KCl, 3 mM MgCl_2_, 0.05 mM dNTPs, 75 U of Ex Taq DNA Polymerase (Diatech Pharmacogenetics, Ancona, Italy), Eva Green 1X (Biotium, Fremont, CA, USA), and 0.3 μM of primers mapping in the flanking regions of exon 2 (5′->3′): Fwd TCATCTGATAGTTTAAGGGCCCG, Rev ATTTCCTAACACTTACCCATGTGC.

Thermal profile was defined as follow: 95 °C 3 min, 40 cycles of 95 °C 30 s, 64 °C 30 s, 72 °C 30 s and a final step of 72 °C 5 min. Exon 2 was analyzed by using the Corbett Rotor-Gene 6000 (Qiagen, Germantown, MD, USA). Data were analyzed using Rotor-Gene Q Series Software 2.3.1 (Qiagen, Germantown, MD, USA).

##### Sanger Sequencing

Sanger Sequencing was performed as previously described [37]. PCR products were analyzed using the Bioanalyzer 2100 (Agilent Technologies, Santa Clara, CA, USA) and purified with the MinElute PCR Purification Kit (Qiagen, Milan, Italy). Sanger Sequencing was done with the BigDye Terminator v1.1 Cycle Sequencing Kit (Thermo Fisher Scientific, Waltham, MA, USA) and the ABI-PRISM 3100 Genetic Analyzer (Thermo Fisher Scientific, Waltham, MA, USA), after sequencing amplification purification using Centrisep Spin Columns (Princeton Separation, Thermo Fisher Scientific, Waltham, MA, USA), all in accordance with operational manuals.

### 4.3. Albumin Treatment of Podocyte Cell Clones

Previous stimulation experiments with albumin (BSA; Sigma-Aldrich, St. Louis, MO, USA) on cultured human podocytes, performed using a concentration range of 10 μg/mL–30 mg/mL [20], showed that ClC-5 was clearly upregulated at both the mRNA and protein levels by 10 and 30 mg/mL BSA. Therefore, these two concentrations of albumin were used for the experiments. Mutant (*CLCN5*^−/−^) and WT clones were plated and cultured in 6 or 96-well tissue culture plates (Falcon), for RT-PCR and ICW analysis respectively. Cells were cultured in serum deprivation (1%) starting from 24 h before stimulation, and then incubated with 10 and 30 mg/mL of BSA for 24, 48, and 72 h. Mutant and WT cells were cultured without stimulation as controls (CTR). Each stimulation experiment was run twice, carrying out two independent experiments. For each experiment, treatments were performed in triplicate for qRT-PCR analysis, and in quadruplicate for ICW experiments.

### 4.4. Quantitative Analysis of Clones’ mRNA

The RNeasy Mini Kit (Qiagen, Hilden, Germany) was used to isolate total RNA. RNA was quantified with the NanodropONE spectrophotometer. RNA purity was checked from the A260/A280 ratio and its integrity was tested by capillary electrophoresis on the Agilent RNA Nano chip using the Agilent 2100 Bioanalyzer (Agilent Technology, Santa Clara, CA, USA). Only RNA with an RNA integrity number of at least of 9 was used for Real-Time PCR analyses.

#### 4.4.1. Reverse Transcription

RNA was retro-transcribed from a starting quantity of 100 ng in a final volume of 20 µL. The reaction mix was prepared as follows: 5 mM MgCl_2_; 1 mM dNTPs; 2.5 µM random hexamers; 1 U RNase inhibitor; 2.5 U MuLV reverse transcriptase (Thermo Fisher Scientific, Waltham, MA, USA) in 50 mM KCl, 10 mM Tris HCl pH 8.3. Reactions were performed on the 2720 thermal cycler (Thermo Fisher Scientific, Waltham, MA, USA) applying the following thermal profile: RT for 10 min, 42 °C for 30 min, 65 °C for 5 min, 4 °C for 5 min.

#### 4.4.2. Real Time PCR

Primer pairs for the region of interest were designed according to stringent parameters to ensure successful assays and a convenient experimental design by using Primer3 software ver. 4.0 (http://primer3.ut.ee, accessed on 22 November 2022). The NCBI Primer-BLAST program was used for in silico specificity analysis (www.ncbi.nlm.nih.gov/tools/primer-blast/index.cgi, accessed on 22 November 2022). Amplification curves were established for all primers and resulted in efficiencies of at least 85%. Primers used are listed in Table 3. For each fragment to analyze, 1 µL of cDNA was amplified in a 20 µL final volume of reaction mix using SYBR Green Master Mix (EurX, Gdansk, Poland) according to the manufacturer’s instructions. Reactions were performed on the RotorGene (Corbett Research Qiagen, Hilden, Germany). Appropriate primer dilutions and annealing temperatures are given in Table 3. Data were analyzed using the ΔΔCt method, normalizing on two different housekeeping genes (glyceraldehyde 3-phosphate dehydrogenase (*GAPDH*) and hypoxanthine guanine phosphoribosyl transferase (*HPRT1*), according to the MIQE guidelines [38]. Microchip electrophoresis on the Agilent 2100 BioAnalyzer, Sanger sequencing, and melting curve analysis were used to check the specificity of the PCR products.

### 4.5. Qualitative Analysis of CLCN5 mRNA

To analyze gene expression of the canonical *CLCN5* isoform (*CLCN5*-204 Transcript ID ENST00000376108) and of the isoform that lacks the 5′ exons and encoding the shorter ClC-5 protein of 590 aa (*CLCN5*-206 ID transcript ESNT 000006422383.1), three sets of primers were designed. The nucleotide composition of primers and the length of amplified products are reported in Table 4. One set is constituted by a 5′-end specific forward primer, i.e., located in exon 1 (1F), and the reverse in exon 4 (4R), encompassing exon 2. The second set of primers comprises a forward primer located in exon 8 (8F) and the reverse in exon 10 (10R). The third set was designed to specifically amplify *CLCN5*-206 isoform, with the forward located in exon 8 (exon 3 of isoform *CLCN5*-206, nested 206F) and the reverse in the sequence that is only present in the *CLCN5*-206 isoform (exon 4, nested 206R) (Figure 13).

An aliquot of 2 μL of cDNA was used to amplify canonical isoform in a final volume of 25 μL containing 0.2 mM dNTPs, 0.4 μM of each primer, 0.04 U JumpStart Taq (Sigma-Aldrich, St Louis, MO, USA) in 50 mM KCl, and 10 mM Tris HCl pH 8.3. The amplification profile for each primer set consisted of an initial denaturation at 95 °C for 5 min, followed by 34 amplification cycles (45 s at 94 °C, 45 s at specific Ta °C, 1 min at 72 °C), and an extension at 72 °C for 7 min. PCR amplification products were visualized using the Agilent 2100 Bioanalyzer.

To enhance the sensitivity and specificity of DNA amplification regarding isoform *CLCN5*-206, nested PCR was used. In this method, two PCRs were applied using two different sets of primers. In the first PCR, a long region was amplified using two primers specific to the outer region of the target DNA sequence that amplify the entirety of the target sequence to yield a primary amplicon (forward primer 8F and reverse 10R) (Table 4 and Figure 13). After the successful amplification was confirmed by Agilent 2100 Bioanalyzer, a second round of nested PCR amplification was done. A small aliquot of the primary PCR amplification products was used as template for the second PCR. The second PCR amplification was performed using the internal pair set of primers designed to specifically amplify *CLCN5*-206 isoform: a forward primer nested 206F and the reverse nested 206R (Table 4 and Figure 13). The correct amplificon was confirmed by Agilent 2100 Bioanalyzer and Sanger sequencing. Three independent amplifications from the first round of PCR for each clone were performed.

### 4.6. Protein Expression in Podocyte Cell Clones

#### 4.6.1. In-Cell Western (ICW)

Protein expression was assessed in mature podocyte clones cultivated on 96-well plates, fixed after specific experimental treatment with cold methanol for 10 minutes at RT. Cells were washed five times with 0.1% Triton X-100 in PBS, then blocked in a blocking solution of 5% milk in 0.1% Triton X-100 in PBS for 40 min at RT with moderate shaking. Samples were incubated with primary antibody (Table 2) diluted in the same medium at 4 °C overnight in a humidified chamber. Then, five washes with 0.1% Triton-X 100 in PBS washing solution were carried out. Secondary antibodies were diluted in blocking solution, as reported in Table 2, and incubated for 1 h at RT with gentle shaking. The intensity of the labelled proteins was measured using the Odyssey CLx imaging system (LI-COR, Lincoln, NE, USA). Negative controls were run by omitting primary antibody. Background values were obtained by omitting primary and secondary antibodies. Signals were normalized for the amount of cells measured by methylene blue staining, as described elsewhere [39,40]. Briefly, cells were stained for 30 min with 1% methylene blue in 0.01 M borate buffer, pH 8.5. After repeated washing with borate buffer, the fixed stain was eluted with 0.1 N HCl/ethanol 1:1 (*vol/vol*). Absorbance was measured at 650 nm with the 680 Microplate Reader (Bio-Rad, Hercules, CA, USA).

#### 4.6.2. F-Actin Cytoskeleton Staining

Cells were chemically fixed on slides using a 4% para-formaldehyde solution in PBS. F-actin was stained using Phalloidin-iFluor-647 (ab176759, Abcam, Cambridge, UK) at 1:1000 dilution for 1 h. Nuclear counterstain was performed using 40,6-diamidin-2-phenylindol (DAPI, Vector Laboratories, Burlingame, CA, USA), at 1:1000 dilution. After staining, fixed samples were imaged by a DMI6000CS-TCS SP8 fluorescence microscope (Leica Microsystems, Wetzlar, Germany) with a 20×/0.4 objective using a DFC365FX camera (Leica Microsystems, Wetzlar, Germany) and analyzed with the LAS-AF software 3.1.1 (Leica Microsystems, Wetzlar, Germany).

### 4.7. Cell Viability Assessment

Cells were cultured in 96-well tissue culture plates and allowed to grow. Viability was assessed at different times of albumin treatment (48 or 72 h) and at different concentrations (range 10 µg/mL–30 mg/mL) by colorimetric assays [39,40]. Cells were fixed with cold methanol for 10 min, then stained with 1% methylene blue in 0.01 M borate buffer (pH 8.5) for 30 min. After repeated washing, the unbound staining solution was eluted with a 1:1 mixture of ethanol and 0.1 N HCl, and read at an absorbance of 650 nm. Methylene blue only stains cells attached to the substrate before fixation (i.e., living cells) and, thus, quantifies their viability.

### 4.8. Statistical Analysis

Statistical analysis was performed using a non-parametric test (Mann–Whitney U test) due to the small sample size. Statistical significance was assessed using R software v.4.1.0 [41]. Results with *p*-values below 0.05 were considered significant.

## Figures and Tables

**Figure 1 ijms-24-01313-f001:**
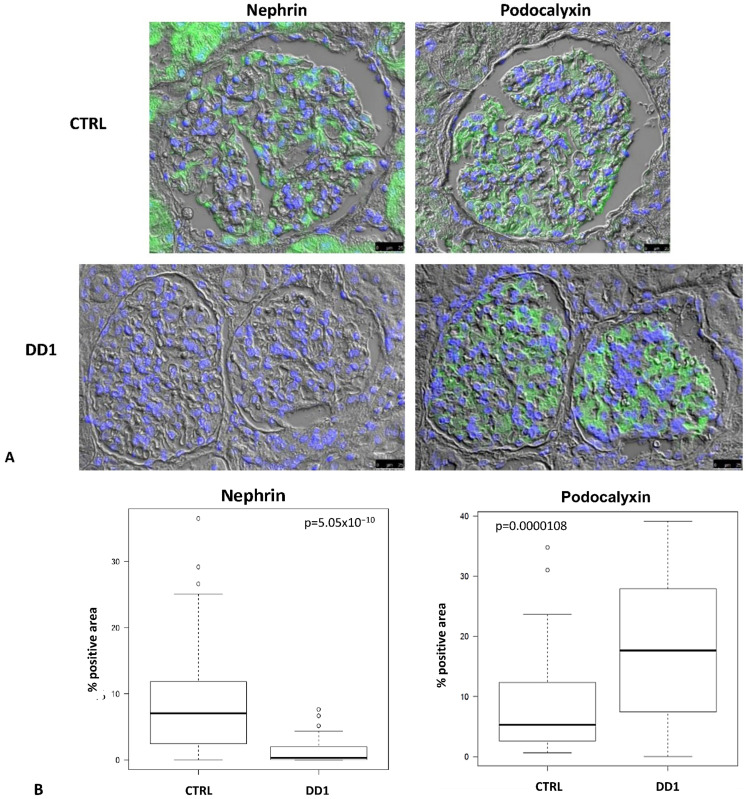
Immunofluorescence (IF) in serial sections of glomeruli showing the expression pattern of nephrin and podocalyxin in control (CTRL) and Dent Disease 1 (DD1) kidneys biopsies. (**A**): Representative image disclosing the absence of nephrin positivity in DD1 glomeruli while podocalyxin immunolabeling was present in both CTRL and DD1 glomeruli. Green: Nephrin; Blue: DAPI. Images were acquired using a DMI6000CS-TCS SP8 fluorescence microscope (Leica Microystems, Wetzlar, Germany) with 20×/0.4 objective. Scale bar 25 μm. (**B**): Morphometric evaluation of nephrin and podocalixin IF in CTRL and DD1 glomeruli. Boxplots show IF staining scores (% positive area).

**Figure 2 ijms-24-01313-f002:**
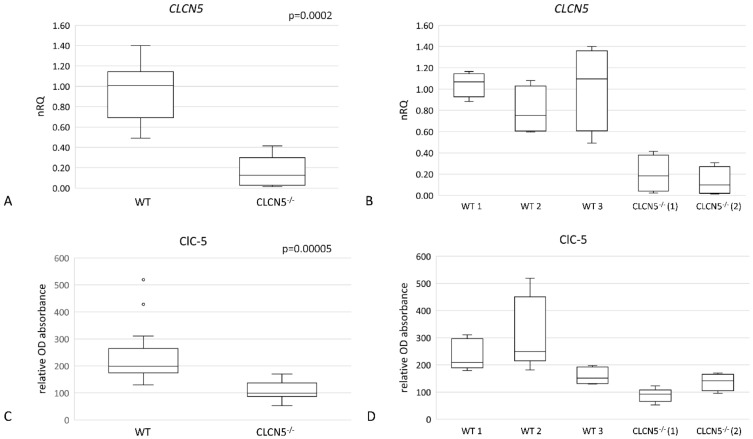
*CLCN5* gene and protein expression in WT and *CLCN5*^−/−^ clones. Boxplots show (**A**,**B**) relative mRNA, as determined by qRT-PCR and (**C**,**D**) protein expression, as determined by ICW analysis. *p*-values were obtained with the Mann–Whitney U test. Results are from two independent experiments performed in triplicate. Abbreviations: nRQ: normalized relative quantity; OD, optical density.

**Figure 3 ijms-24-01313-f003:**
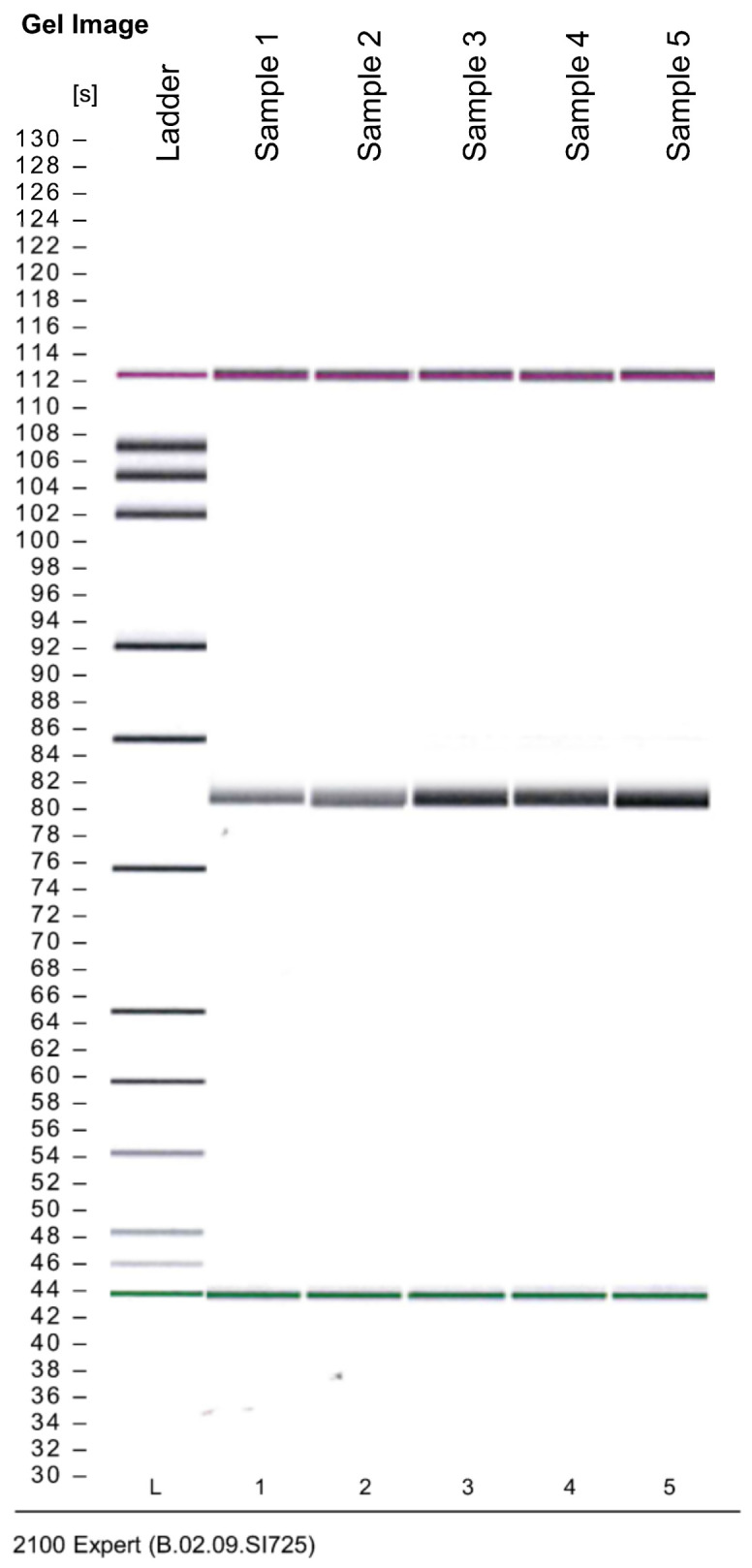
Qualitative RT-PCR analysis of canonical *CLCN5*-204 isoform mRNA. Image generated by Agilent 2100 bioanalyzer of amplified exon 2 in *CLCN5^−/−^* (lines 1 and 2) and WT clones (lines 3, 4, and 5).

**Figure 4 ijms-24-01313-f004:**
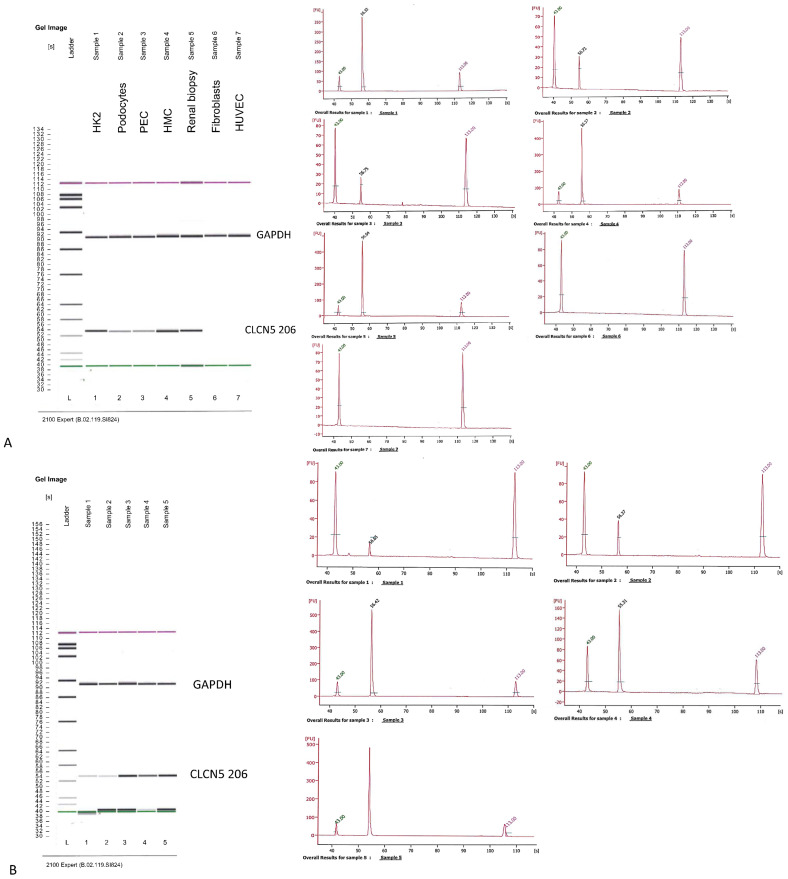
Qualitative RT-PCR analysis of *CLCN5*-206 isoform mRNA. Images generated by Agilent 2100 bioanalyzer of amplified housekeeping gene (glyceraldehyde 3-phosphate dehydrogenase—*GAPDH*) and *CLCN5*-206 isoform in (**A**) different human kidney cell lines, and (**B**) in *CLCN5^−/−^* (Lines 1 and 2) and WT clones (Lines 3, 4, and 5). Chromatograms refer to *CLCN5*-206 isoform amplicons. Images are representative of three independent experiments.

**Figure 5 ijms-24-01313-f005:**
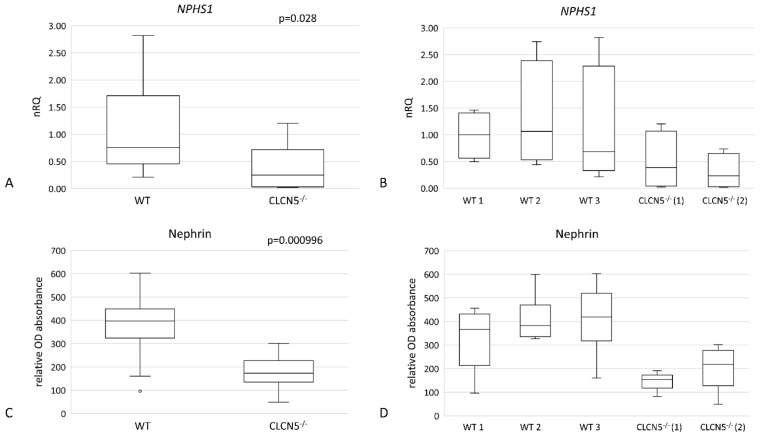
Nephrin gene (*NHPS1*) and protein expression in WT and *CLCN5*^−/−^ clones. Boxplots show (**A**,**B**) relative mRNA, as determined by qRT-PCR, and (**C**,**D**) protein expression, as determined by ICW. *p*-values were obtained with the Mann-Whitney U test. Results are from two independent experiments performed in triplicate. Abbreviations: nRQ: normalized relative quantity; OD, optical density.

**Figure 6 ijms-24-01313-f006:**
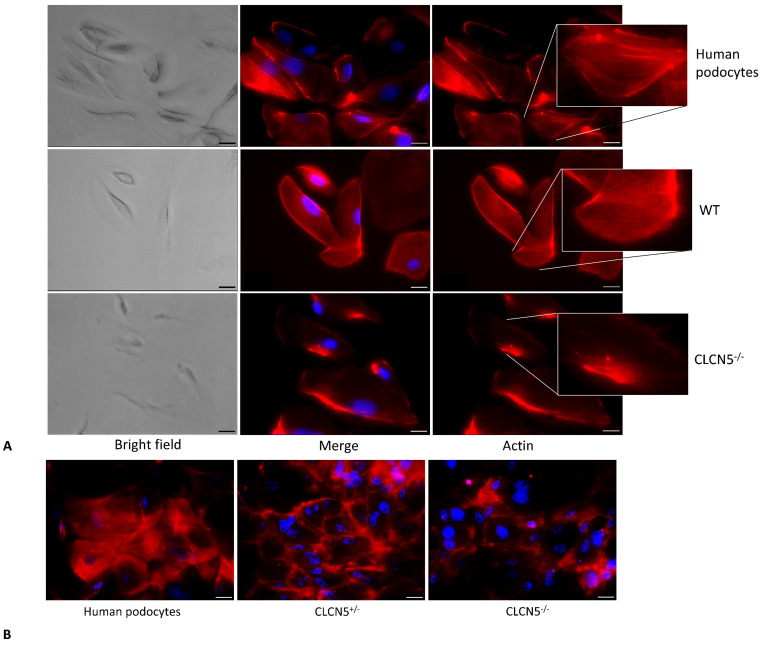
Phalloidin fluorescence labelling of F-actin of human podocytes, WT, and *CLCN5^−/−^* clones. Representative images (with boxed photos showing zoomed-in area) demonstrating: (**A**) no difference in the staining pattern between human podocytes and WT clones, while irregular F-actin distribution in *CLCN5*^−/−^ clones was markedly present; (**B**) differences in the staining pattern between heterozygous (*CLCN5*^+/−^) and homozygous (*CLCN5*^−/−^) clones: actin cytoskeleton alteration of heterozygous clones was intermediate between normal human podocytes and homozygous clones. Red: actin; Blue: DAPI. Images were acquired using a DMI6000CS-TCS SP8 fluorescence microscope (Leica Microystems, Wetzlar, Germany) with 20×/0.4 objective. Fluorescence microscope images are representative of three independent experiments. Merge: merge with DAPI. Scale bar 25 μm.

**Figure 7 ijms-24-01313-f007:**
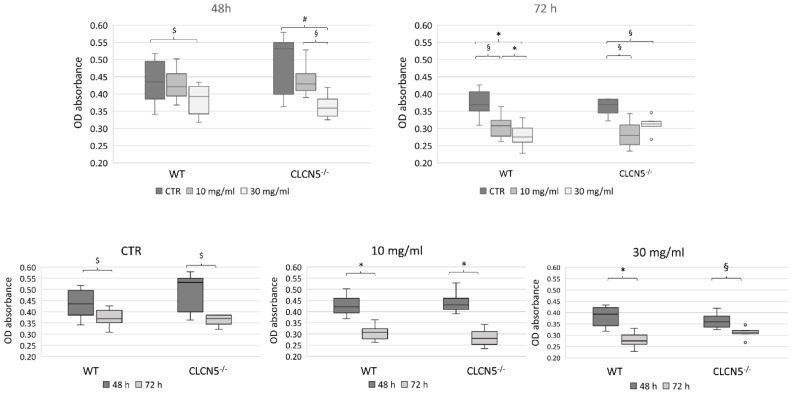
Cell viability assessment. Results of methylene blue assay of WT and *CLCN5*^−/−^ cells treated with albumin (range 10 µg/mL–30 mg/mL) for 48 and 72 h. *p*-values were obtained with the Mann-Whitney U test. The results presented are from two independent experiments performed in triplicate. * *p* < 0.001; § *p* < 0.005; # *p* < 0.01; $ *p* < 0.05. OD: optical density. CTR: unstimulated cells.

**Figure 8 ijms-24-01313-f008:**
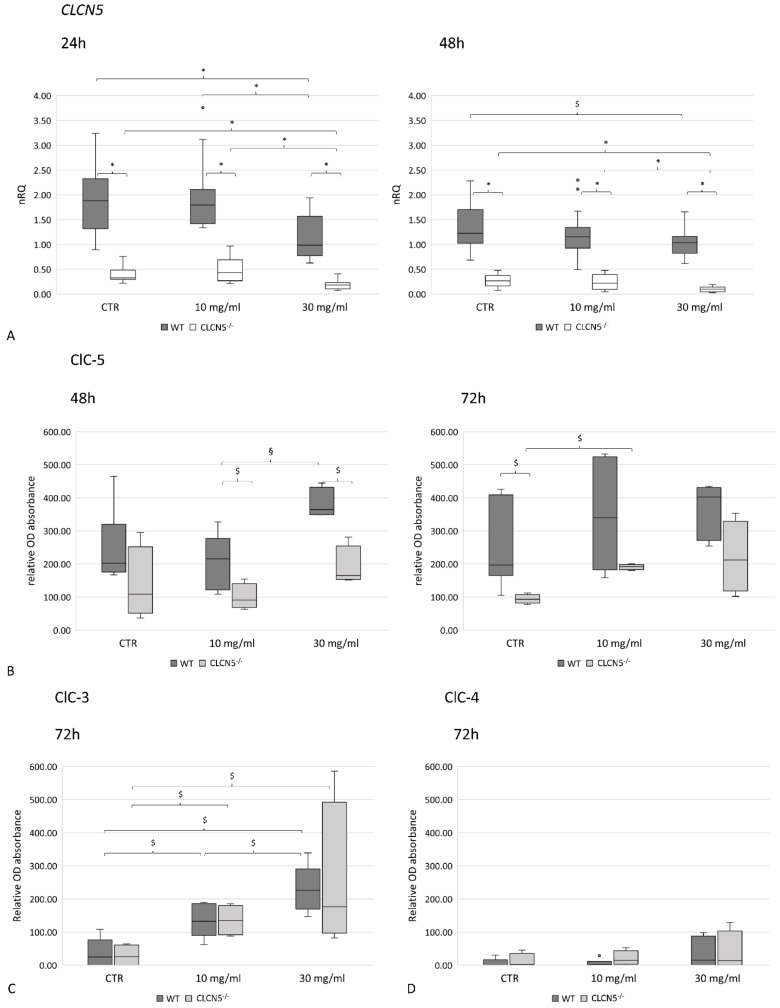
ClC-5, ClC-3, and ClC-4 expression in control WT and *CLCN5*^−/−^ clones following albumin treatment. Boxplots shows medians and interquartile ranges of (**A**) relative *CLCN5* mRNA, as determined by qRT-PCR, and ClC-5 (**B**), ClC-3 (**C**), and ClC-4 (**D**) protein expression as determined by ICW. *p*-values were obtained with the Mann–Whitney U test. * *p* < 0.001; § *p* < 0.005; $ *p* < 0.05. Results are from two independent experiments performed in triplicate (qRT-PCR) and in quadruplicate (ICW). Abbreviations: nRQ: normalized relative quantity; OD, optical density, CTR: unstimulated cells.

**Figure 9 ijms-24-01313-f009:**
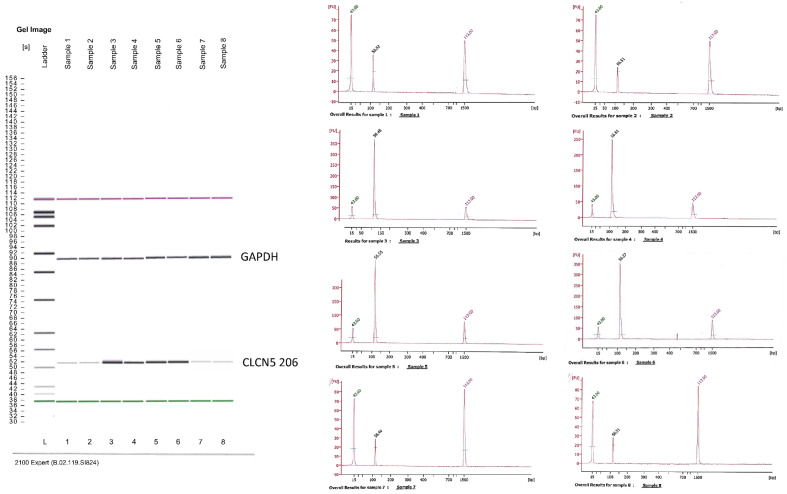
Qualitative RT-PCR analysis of the *CLCN5*-206 isoform mRNA. Image generated by the Agilent 2100 bioanalyzer of amplified housekeeping gene (glyceraldehyde 3-phosphate dehydrogenase—*GAPDH*) and *CLCN5*-206 isoform in *CLCN5^−/−^* (Lines 1 to 4: lines 1–2 CTR, lines 3–4 30 mg/mL of albumin after 48 h of exposure) and WT clones (Lines 5 to 8: lines 5–6 CTRL, lines 7–8 30 mg/mL of albumin after 48 h of exposure). Chromatograms refer to *CLCN5*-206 isoform amplicons. Image is representative of the results of three independent experiments.

**Figure 10 ijms-24-01313-f010:**
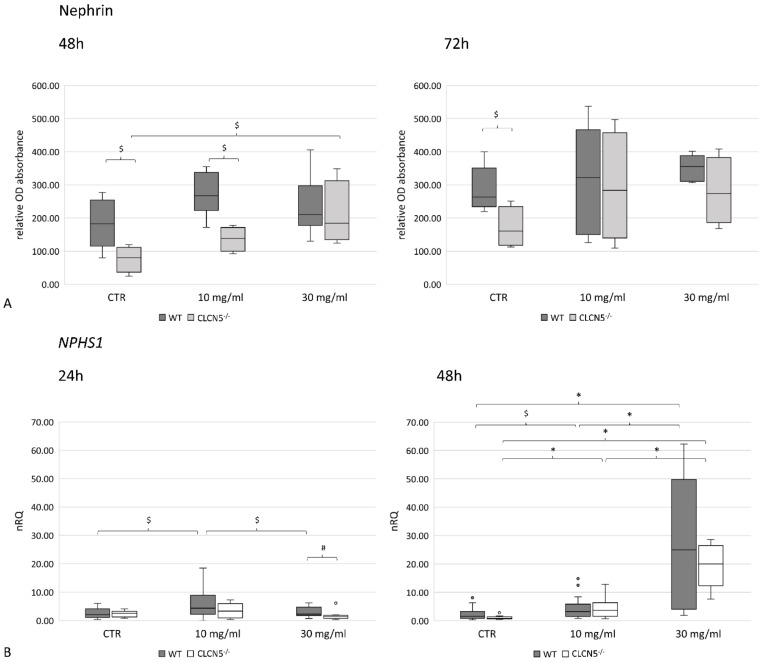
Nephrin expression in WT and *CLCN5*^−/−^ clones following albumin treatment (10 mg/mL and 30 mg/mL) at different time points (24, 48, and 72 h). Boxplots show medians and interquartile ranges of (**A**) protein expression, as determined by ICW and (**B**) relative mRNA, as determined by qRT-PCR. *p*-values were obtained with the Mann–Whitney U test. * *p* < 0.001; # *p* < 0.01; $ *p* < 0.05. Results are from two independent experiments performed in triplicate (RT-PCR) and in quadruplicate (ICW). Abbreviations: nRQ: normalized relative quantity; OD, optical density.

**Figure 11 ijms-24-01313-f011:**
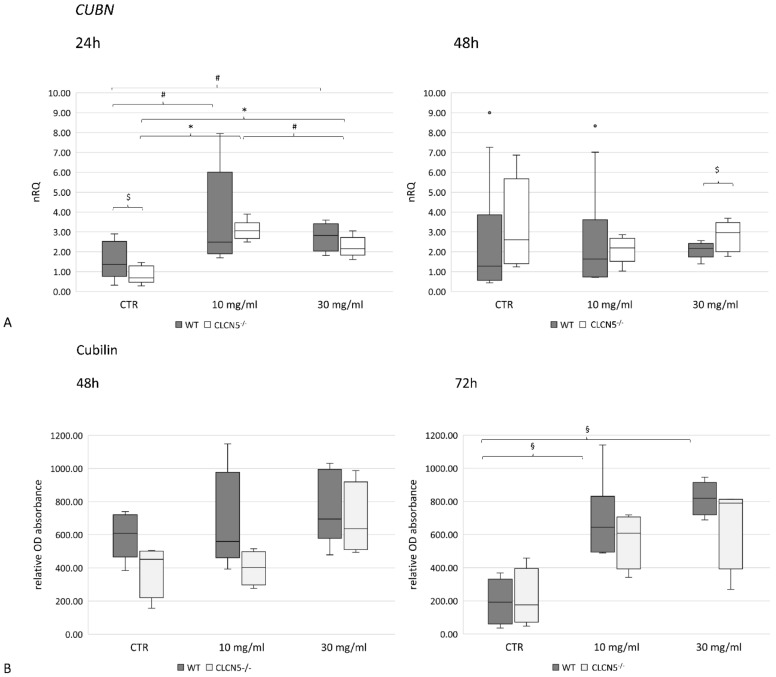
Cubilin expression in WT and *CLCN5*^−/−^ clones following albumin treatment (10 mg/mL and 30 mg/mL) at different time points (24, 48, and 72 h). Boxplots show medians and interquartile ranges of (**A**) relative mRNA, as determined by qRT-PCR, and (**B**) protein expression, as determined by ICW. *p*-values were obtained with the Mann–Whitney U test. * *p* < 0.001; § *p* < 0.005; # *p* < 0.01; $ *p* < 0.05. Results are from two independent experiments performed in triplicate (qRT-PCR) and in quadruplicate (ICW). Abbreviations: nRQ: normalized relative quantity; OD, optical density.

**Figure 12 ijms-24-01313-f012:**
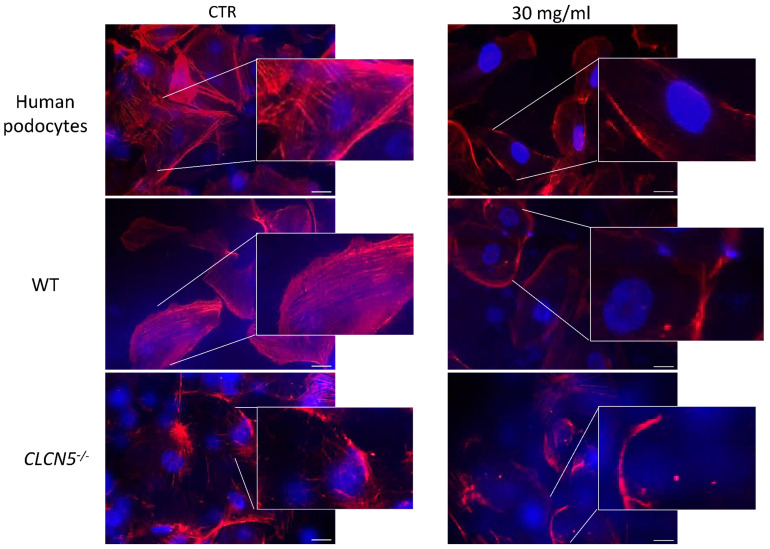
Phalloidin fluorescence labelling of F-actin of human podocytes, WT (clones subjected to CRISPR/Cas9 editing but without mutation), and mutant clones (*CLCN5*^−/−^) after exposure to albumin 30 mg/mL for 72 h. Representative images with boxed photos showing zoomed-in area demonstrating no differences in the F-actin staining pattern of *CLCN5*^−/−^ mutant between CTR and albumin treatment. Alterations of F-staining pattern in WT clones and in human podocytes under albumin treatment were observed. Red: actin; Blue: DAPI. Images were acquired using a DMI6000CS-TCS SP8 fluorescence microscope (Leica Microystems, Wetzlar, Germany) with 20×/0.4 objective. Fluorescence microscope images are representative of three independent experiments. Scale bar 25 μm.

**Figure 13 ijms-24-01313-f013:**
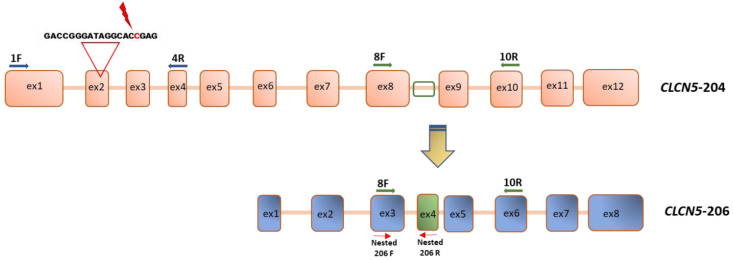
Schematic representation of primer localization in *CLCN5* cDNA. Primer pairs 1F-4R amplify canonical *CLCN5*-204 isoform; primer pairs nested 206F-nested 206R amplify only *CLCN5*-206 isoform starting from a primary amplicon obtained by a first PCR amplification with primer pairs 8F-10R.

**Table 1 ijms-24-01313-t001:** Summary of histopathological and ultrastructural analyses of the three DD1 biopsies and the four control biopsies. CTRL: control biopsies TEM: transmission electron microscopy. N/A: not applicable.

DD1	ClC-5 Mutation	Age at Biopsy (Years)	Indication for Biopsy	Proteinuria	Number of Glomeruli Examined	Number of Glomeruli with Global Sclerosis	Podocytes Structure (TEM)
**a**	**p.(Q600*)**	6	proteinuria	1.2 g/day	18	5	foot process effacement
**b**	**p.(R34*)**	11	Nephrotic syndrome	1.6 g/day	16	8	foot process effacement
**c**	**p.(V308M)**	9	proteinuria and hematuria	0.5 g/day	29	1	normal
**CTRL** **(n = 4)**	N/A	55 (range 41–67)	N/A	N/A	47	0	N/A

**Table 2 ijms-24-01313-t002:** Antibodies used in IHC, IF, and ICW experiments.

	Target	Clone	Host	Manufacturer	Code	Conjugation	Dilution IHC	Dilution IF	Dilution ICW
**primary antibody**	ClC-5	-	rabbit	ATLAS ANTIBODY	HPA000401		1:200		1:200
	ClC-3	K-17	goat	Santa Cruz Biotechnology	sc-17572				1:50
	ClC-4	-	rabbit	ATLAS ANTIBODY	HPA063637				1:150
	Cubilin	-	sheep	R&D Systems	AF3700				1:200
	Nephrin	-	guinea pig	Progen	GP-N2			1:25	1:25
	Podocalyxin	4F10	mouse	Santa Cruz Biothecnology	sc-23903			1:100	
**IHC**	anti-rabbit	-	-	Dako	K4002	EnVision + System-HRP Labeled Polymer	Reedy to use		
**IF secondary** **antibody**	Anti-rabbit	-	donkey	Santa Cruz Biotechnology	sc-362291	CFL 647		1:100	
	Anti-guinea pig		donkey	Jackson ImmunoResearch laboratory	706-545-148	Alexa 488		1:100	
	Anti-mouse		goat	ThermoFisher Scientific	A-11001	Alexa 488		1:1000	
**ICW secondary antibody**	Anti-rabbit	-	donkey	LI-COR	926-32213	IRDye 800CW			1:800
	Anti-sheep	-	donkey	ThermoFisher Scientific	A-21102	Alexa Fluor 680			1:1000
	Anti-goat	-	donkey	LI-COR	926-32214	IRDye 800CW			1:800
	Anti-guinea pig	-	donkey	LI-COR	925-32411	IRDye 800CW			1:600

**Table 3 ijms-24-01313-t003:** Primer used in Real Time PCR analyses and Real Time PCR amplification conditions.

Name	NCBI Reference Sequence	Sequence (5′-3′)	[PRIMER] µM	T_a_ (°C)	Size (bp)	Efficiency (%)
*GAPDH* For	NM_17851.1	GAAGGTGAAGGTCGGAGT	0.4	62	92	96
*GAPDH* Rev		TGGCAACAATATCCACTTTACCA	0.4			
*HPRT1* For	NM_000194.2	CCTGGCGTCGTGATTAGTGA	0.4	62	140	86
*HPRT1* Rev		TCTCGAGCAAGACGTTCAGT	0.4			
*CLCN5* For	NM_000084.4	TGCTGGAACTCTGAGCATGT	0.2	64	162	99
*CLCN5* Rev		TACGGCAAGGAAGGCAAATA	0.2			
*CLCN3* For	NM_001829.4	TGGAGCAGGTGTTATTATGGAC	0.4	62	105	94
*CLCN3* Rev		ATGCTGCCTCCATTTGTCAT	0.4			
*CLCN4* For	NM_001830.4	GTCGCGCTGAAGAAAGGAT	0.4	62	122	89
*CLCN4* Rev		TCAGGTTTCCAGAGCCACTC	0.4			
*CUBN* For	NM_001081.3	GCCGTGAGAAAGGATTTCAG	0.4	62	118	85
*CUBN* Rev		TCCTTGTTTGGTGGATACCTG	0.4			
*NPHS1* For	NM_004646.3	CAACTGGGAGAGACTGGGAGAA	0.2	64	189	87
*NPHS1* Rev		AATCTGACAACAAGACGGAGCA	0.2			

**Table 4 ijms-24-01313-t004:** Primers used for RT/PCR analysis of the *CLCN5* mRNA.

	Name	NCBI Reference Sequence	Sequence (5′-3′)	[PRIMER] µM	T_a_ (°C)	Size (bp)
**Canonical isoform** ***CLCN5*-204**	**1F**	NM_000084.4	AAGCTCCCCAACCTGAATGA	20	56	360
	**4R**		TGTCTATCAAACCAGCTAACGA	20		
**Isoform** ***CLCN5*-206**	**8F**		CAGCCATCACTGCCATCCTG	20	64	630
	**10R**		CATCTATGATGCCCACATCCGT	20		
	**nested 206F**		GGCAGCTGGTTTAACACTC	20	60	110
	**nested 206R**		AGCCTGAACTCTCCAGACCA	20		

## Data Availability

Data is contained within the article or supplementary material.

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
