# Peer review of "Emerging Perspectives on the Rare Tubulopathy Dent Disease: Is Glomerular Damage a Direct Consequence of ClC-5 Dysfunction?"

_ijms, 2023, doi:10.3390/ijms24021313_

Round 1
Reviewer 1 Report
1. For the figure 4 and its result, how did the authors get the conclusion that" podocytes and PECs exhibited the lowest signal"? How many samples did you have and any quantification data?
2. For figure 6 and 12, negative control staining should be included.
3. In the result and method sections, authors said they used a range of concentrations of albumin (range 10 μg/ml–30 mg/ml) at different time points (24, 48, and 72 hours). But why did the authors only show 10 mg/ml and 30mg/ml??
Author Response
- In figure 4, we showed the results of qualitative analysis of PCR products that were generated through a nested PCR as described in the paragraph “Qualitative analysis of CLCN5 mRNA” of Material and Methods section (page 25, line 653). A nested PCR could not be really quantitative for its intrinsic nature, as commented in the results section (at page 14, line 316), but it is of value for examining the presence of very rare mRNAs. With this approach, we found that CLCN5-206 isoform is present in kidney tissue, specifically in some cell types of the nephron (as reported at page 6, line 180-182). We tentatively gave a measure of CLCN5-206 expression by judging the band intensity from the Agilent Bioanalyser patterns of amplicons. As you can see, the band intensity differs among the samples, giving that the intensity of housekeeping gene amplicon was nearly the same for all samples. This result was also confirmed by Agilent Bioanalyser chromatograms that measure the quantity of each amplicon. We have added chromatograms to figure 4 and clarified the issue by adding the following to the text at pag 6, line 184-188: “Although nested PCR cannot be quantitative for its intrinsic nature, we tried to give a measure of CLCN5-206 expression by judging band intensity from the Agilent pattern of amplicons. The PCR products for the housekeeping gene, obtained from the first round of nested PCR, assured us that the starting conditions for the second round of PCR were quite similar for all samples.”
- Figures 6 and 12 show images taken from cells fixed on slides and simply stained using the phalloidin coniugated to a fluorescence dye. Correctly designed fluorescent phalloidins only binding to the native quaternary structure of F-actin and therefore exhibit negligible non-specific staining, resulting in a clear and distinct contrast between stained and unstained areas. This can be clearly appreciated from the images of figure 6 and 12. For the reason reported above, we did not perform negative control, i.e. not stained slides. We have added the specifics of the protocol used in Materials and Methods section, paragraph “F-actin cytoskeleton staining” (page 27, line 710).
- We thank the reviewer for pointing out this discrepancy. Indeed, previous BSA stimulation experiments on cultured human podocytes were performed using a range 10 ug-30mg/ml (Gianesello L, Priante G, Ceol M, Radu CM, Saleem MA, Simioni P, Terrin L, Anglani F, Del Prete D. Albumin uptake in human podocytes: a possible role for the cubilin-amnionless (CUBAM) complex. Sci Rep. 2017 Oct 20;7(1):13705. doi: 10.1038/s41598-017-13789-z.). They showed that ClC-5 was clearly upregulated at both mRNA and protein level by 10 and 30 mg of BSA. Therefore, we used only these two concentrations in this study. We added the following in Materials and Methods section paragraph “Albumin treatment of podocyte cell clones” page 24, lines 608-612: “Previous stimulation experiments with BSA on cultured human podocytes, performed using a concentration range 10 ug/ml-30mg/ml (20), showed that ClC-5 was clearly upregulated at both the mRNA and protein level by 10 and 30 mg/ml BSA. Therefore, these two concentrations of albumin were used for the experiments” and we corrected the incorrect information in the entire manuscript.
Reviewer 2 Report
The authors evaluated the expression of podocyte ClC-5 and the involvement of podocyte CIC-5 loss in the pathophysiology of dent disease. This study could provide some insights for the scientific community and the practitioners.
However, there are several issues in this study and this manuscript:
Major points:
1. The authors should provide in table 1 the characteristic of each control individual from whom the biopsies were obtained.
2. For data set corresponding to Figure 2, Figure 3 and Figure 5, the authors may consider doing more independent experiments to obtain more convincing data.
3. For Figure 3, 4 and 9, the authors should also provide the quantitative data.
4. The authors should consider providing the in vivo cytoskeleton data obtained from human DD1 patients in Figure 6.
5. For Figure 6 and 12, the authors should also provide boxed photos showing zoomed-in areas, and the quantitative data.
Minor points:
1. In Figure 7, the authors should also consider providing the 24 h data set.
To summarize, for this manuscript, the authors need to address the essential issues mentioned above. I, therefore recommend reconsideration after major revision.
Author Response
- As described in “Renal biopsies” paragraph of the Materials and Methods section, the control biopsies were four cortical tissues obtained from surgical biopsies of nephrectomies for renal cancer (sites remote from the tumor-bearing renal tissue). We performed Hematoxylin/Eosin staining and immunofluorescence to exclude altered morphology of glomeruli. These specimens are considered classical control renal tissues for renal biopsy studies of patients with nephropathy, and, as such, they are not subjected to extensive diagnostic analysis such as TEM. In table 1, we have reported the information you suggested but summarized.
- Figure 3 provides only qualitative data from the experiment we performed to analyze whether exon 2 skipping occurred during mRNA processing. It is shown that by amplifying the region of CLCN5 canonical mRNA spanning exon 2 where the mutation is located, no band smaller than expected is present in mutant clones compared with WT clones. We added this specification to the text at page 6, line 170-172: “We did not identify smaller PCR products than expected in mutant clones compared with WT clones, thus indicating that no alteration in mRNA processing occurred”. Furthermore, we added the following in Materials and Methods section paragraph “Qualitative analysis of CLCN5 mRNA” (page 26, lines 684-685): “Three independent amplifications from the first round of PCR for each clone were performed.”
Figures 2 and 5 show that ClC5 and Nephrin are significantly under-expressed in the mutant clones. For evaluating the basal expression of ClC-5 and Nephrin, two independent experiments were performed. In each experiment, clones were plated in triplicate on 6 or 96-well plates for RT-PCR and In Cell Western (ICW) analysis respectively. This information was added to “CRISPR/Cas9-mediated genome editing” paragraph of Materials and Methods (page 23, lines 581-584). Considering that two mutant clones and three WT clones were analyzed, it is reasonable to assume that the data obtained are reliable.
- Figure 3 provides only qualitative data from the experiment we performed to analyze whether exon 2 skipping occurred during mRNA processing.
Indeed, as specified in the figure legends, Figures 4 and 9 show the results of qualitative analysis of PCR products that were generated through a nested PCR as described in the paragraph “Qualitative analysis of CLCN5 mRNA” of the Material and Methods section (page 25). A nested PCR could not be quantitative for its intrinsic nature, as commented in the results section (at page 14, line 31680 but it is of value for examining the presence of vary rare mRNAs. With this approach, we found that CLCN5-206 isoform is present in kidney tissue, specifically in some cell types of the nephron (as reported at page 6, line 180-182). We tentatively gave a measure of CLCN5-206 expression by judging the band intensity from the Agilent pattern of amplicons. As you can see, the band intensity differs among the samples, giving that the intensity of housekeeping gene amplicon was nearly the same for all samples. This is also confirmed by Agilent chromatograms that give an estimate of the quantity of each amplicon. We have added chromatograms to figure 4 and 9, and clarified the issue by adding the following to the text at pag 6, line 184-188: “Although nested PCR cannot be quantitative for its intrinsic nature, we tried to give a measure of CLCN5-206 expression by judging band intensity from the Agilent pattern of amplicons. The PCR products for the housekeeping gene, obtained from the first round of nested PCR, assured us that the starting conditions for the second round of PCR were quite similar for all samples.”, and at page 26, lines 684-68: “Three independent amplifications from the first round of PCR for each clones were performed.”.
- Thanks for your suggestion. It should be very interesting to analyze F-actin in DD1 glomeruli. However, there are some technical difficulties for providing in vivo cytoskeleton data. Kidney biopsies from DD1 patients are difficult to collect since a few patients with this rare disease undergo to this diagnostic approach. Add to this the fact that the patients whose results we show in this study are from other nephrology centres, so we obtained a limited number of slides for each patients. Moreover, the few data from the literature show that the F-actin staining with fluorescent phalloidin would require an artificial expansion of the glomeruli for being interpreted correctly (Cortes P, Méndez M, Riser BL, Guérin CJ, Rodríguez-Barbero A, Hassett C, Yee J. F-actin fiber distribution in glomerular cells: structural and functional implications. Kidney Int. 2000 Dec;58(6):2452-61. doi: 10.1046/j.1523-1755.2000.00428.), which is impossible in the case of a human biopsy. As both mesangial and podocytes have stress fibers containing F-actin, the analysis also requires co-localization with podocyte and mesangial markers. For all these reasons, we cannot provide data of F-actin staining in DD1 renal biopsies right now.
- In Figures 6 and 12, we wanted to show the different distribution pattern of F-actin fibers in WT and mutant podocytes, so only qualitative data. We agree with your suggestion to provide inset photos showing the enlarged area for better visualization of the difference in F-actin distribution, which is mainly qualitative as reported in the results section on page 9, line 233-237: “Human podocytes and WT clones showed a pattern of F-actin filaments distributed as bundles of stress fibers along the cell axis, arranged neatly and unbranched, as visualized by fluorescence microscopy. In mutant clones, the orderly arranged stress fibers of the podocyte actin cytoskeleton were disrupted and a marked redistribution of F-actin fibers toward the periphery was observed”.
This type of qualitative analysis was conducted also by Morigi et al (Morigi M, Buelli S, Angioletti S, Zanchi C, Longaretti L, Zoja C, Galbusera M, Gastoldi S, Mundel P, Remuzzi G, Benigni A. In response to protein load podocytes reorganize cytoskeleton and modulate endothelin-1 gene: implication for permselective dysfunction of chronic nephropathies. Am J Pathol. 2005 May;166(5):1309-20. doi: 10.1016/S0002-9440(10)62350-4.).
We agree with you that in somehow the redistribution of stress fibers toward the periphery could be interpreted as a decrease in actin expression, since the most of the cytoplasm turns out to be devoid of F-actin, so quantitative data could help us to understand whether there is also a decrease of F-actin expression in mutant clones. Unfortunately, we cannot provide morphometric data on F-actin staining of clones.
For clarity, we have added the following to the paragraph “CLCN5 down regulation alters podocyte actin cytoskeleton” of the Results section (page 9, line 237-239): “The redistribution of stress fibers toward the periphery could also be interpreted as a decrease in actin expression, since the most of the cytoplasm turns out to be devoid of F-actin.”
Minor points
Because the most important information we derived from our experiments is that related to the level of protein expression, we planned to evaluate the effect of albumin on cell viability at 48 and 72 hours, the times at which the ICW experiments were performed. Therefore, we do not have a data set at 24 hours.
Round 2
Reviewer 1 Report
none
Reviewer 2 Report
In this version, the authors addressed several issues mentioned in the previous comments, or provided somehow reasonable explanations regarding the raised issues. I agree with the authors that at current stage, with these resources, there could still be technical challenges and imperfection in relevant analysis methodology in this type of studies, especially for the human samples from rare disease patients. Considering the overall merit of the manuscript and the fact that the authors have made several modifications to the previous version to address the previously mentioned issues, I recommend acceptance of the manuscript after minor revisions.